# BNT162b2 vaccination enhances interferon-JAK-STAT-regulated antiviral programs in COVID-19 patients infected with the SARS-CoV-2 Beta variant

Ludwig Knabl[1,9✉], Hye Kyung Lee[2,9✉], Manuel Wieser[1], Anna Mur[3], August Zabernigg[3], Ludwig Knabl Sr.[4], Simon Rauch [5], Matthias Bock[5,6], Jana Schumacher[7], Norbert Kaiser[7], Priscilla A. Furth [8✉] & Lothar Hennighausen [2✉]

## Abstract

**Background** SARS-CoV-2 infection activates interferon-controlled signaling pathways and elicits a wide spectrum of immune responses and clinical manifestations in human patients.

**Methods** Here, we investigate the impact of prior vaccination on the innate immune response of hospitalized COVID-19 patients infected with the SARS-CoV-2 Beta variant through RNA sequencing of peripheral blood immune cells. Four patients had received the first dose of BNT162b2 about 11 days prior to the onset of COVID-19 symptoms and five patients were unvaccinated. Patients had received dexamethasone treatment. Immune transcriptomes were obtained at days 7-13, 20-32 and 42-60 after first symptomology.

**Results** RNA-seq reveals an enhanced JAK-STAT-mediated immune transcriptome response at day 10 in vaccinated patients as compared to unvaccinated ones. This increase subsides by day 35. Expression of the gene encoding the antiviral protein oligoadenylate synthetase (OAS) 1, which is inversely correlated with disease severity, and other key antiviral proteins increases in the vaccinated group. We also investigate the immune transcriptome in naïve individuals receiving their first dose of BNT162b2 and identify a gene signature shared with the vaccinated COVID-19 patients.

**Conclusions** Our study demonstrates that RNA-seq can be used to monitor molecular immune responses elicited by the BNT162b2 vaccine, both in naïve individuals and in COVID-19 patients, and it provides a biomarker-based approach to systems vaccinology.

### Plain language summary

The human body's immune response can be triggered by infection or vaccination. Vaccination provides exposure to microbial proteins and trains the immune system in creating a healthy and stronger immune response to subsequent infection. This study looked at changes in the immune response in three groups of individuals, those infected with the Beta SARS-CoV-2 variant following BNT162b2 mRNA vaccination, those infected with Beta SARS-CoV-2 without prior vaccination, and uninfected individuals receiving the BNT162b2 vaccine. It focused on a group of proteins, the JAK-STAT pathway, that plays an important role in the immune response. In previously vaccinated infected individuals JAK-STAT genes were more highly expressed than in unvaccinated infected individuals. JAK-STAT genes were also induced by vaccination alone. This study defines a molecular mechanism for how vaccination strengthens immune responses.

[1] TyrolPath Obrist Brunhuber GMBH, Zams, Austria. [2] National Institute of Diabetes, Digestive and Kidney Diseases, Bethesda, MD 20892, USA. [3] Division of Internal Medicine, Krankenhaus Kufstein, Kufstein, Austria. [4] Krankenhaus St. Vinzenz, Zams, Austria. [5] Division of Anesthesia and Intensive Care Medicine, Krankenhaus Meran, Meran, Italy. [6] Department of Anesthesiology, perioperative Medicine and Intensive Care Medicine, Paracelsus Medical University, Salzburg, Austria. [7] Division of Internal Medicine, Krankenhaus St. Johann, St. Johann, Austria. [8] Departments of Oncology & Medicine, Georgetown University, Washington, DC, USA. [9] These authors contributed equally: Ludwig Knabl, Hye Kyung Lee. ✉email: Ludwig.knabl@tyrolpath.at; hyekyung.lee@nih.gov; paf3@georgetown.edu; lotharh@nih.gov

Our understanding of immune responses following infection with different SARS-CoV-2 variants and at specific stages of COVID-19 disease continues to evolve. While current mRNA-based vaccines have proven very protective of the original SARS-CoV-2 strain, widespread escape of new variants from monoclonal antibody neutralization[1–4] are of concern[5]. Recent data from Qatar demonstrate a BNT162b2 vaccine efficacy of 75% with the Beta (formerly B.1.351) variant[6]. Profiling genomic immune responses of hospitalized COVID-19 patients that had been vaccinated prior to their infection might provide molecular clues on the benefits provided by vaccines in the setting of different SARS-CoV-2 variants.

Type-I interferons (IFNs) are ubiquitously expressed cytokines that control both innate and cell-intrinsic immunity against viral infections[7]. They facilitate antiviral activity and are key immune mediators impacting COVID-19. Impaired type-I IFN signaling may predispose to severe disease and mutations in genes controlling type-I IFN-dependent immunity[8] or the presence of neutralizing auto-antibodies (auto-Abs) against some IFNs[9–11] have been associated with severe COVID-19 disease in at least 13% of patients. Type-I IFNs activate a canonical signaling pathway composed of Janus kinases (JAK) and Signal Transducers and Activators of Transcriptions (STAT) that results in the rapid activation of genetic programs[12]. Interferon-driven transcriptomic responses have been detected in peripheral blood of COVID-19 patients[13] suggesting that transcriptional profiling can elucidate mechanistic underpinnings of inflammatory responses in humans.

Limited data is available on the immune transcriptome response of vaccinated individuals that subsequently contracted COVID-19. The SARS-CoV-2 Beta variant, characterized by the receptor-binding domain (RBD) mutation E484K[14], spread in Tyrol (Austria) in the spring of 2020 and we had access to hospitalized patients, some of which had received the first dose of the BNT162b2 vaccine prior to contracting COVID-19. This "real-world situation" of nine hospitalized patients (four vaccinated and five unvaccinated) provided evidence that prior vaccination led to an enhanced JAK-STAT signature, including the induction of innate immune programs and the *OAS1* gene whose expression is inversely correlated with mortality[15].

Systems vaccinology[16] addresses immune responses to vaccines through genome-scale transcriptome analyses[17,18]. At this point, it is not clear to what extent the specific gene signature identified in our vaccinated COVID-19 cohort relates to vaccine signatures in naïve individuals receiving the first dose of BNT162b2. Towards this end, we performed immune transcriptome analyses of eight naïve individuals that had received the first dose of BNT162b2 and we integrated published data[18]. A unique aspect of this single setting study is that the infected individuals were all exposed to the Beta variant within the same timeframe and vaccinated individuals were immunized within the same timeframe, controlling for environmental and kinetic variables. The vaccination and clinical studies were performed in parallel, all samples were processed by the same scientists, and sequencing was conducted in the same facility, ensuring a well-controlled experimental environment.

Here we demonstrate the transient activation of the JAK-STAT signaling pathway in peripheral immune cells isolated from hospitalized patients infected after the first vaccination. Specifically, increased expression of genes encoding key antiviral proteins is observed. We demonstrate that whole transcriptome investigation from peripheral immune cells is feasible and yields actionable data. Studies covering the intersection of vaccination and COVID-19 provide key information critical for the everyday management of the COVID-19 disease. Future studies in systems vaccinology need to address the protective effect of different vaccines on other SARS-CoV-2 variants, including omicron.

## Methods

**SARS-CoV-2 virus sequencing**. RNA was extracted from the patient's blood using a Maxwell RSC simply RNA Blood purification kit according to the manufacturer's instructions (Promega, USA). Library preparation and sequencing was performed as described[19]. In short, cDNA was obtained by using reverse transcriptase with random priming. Following cDNA synthesis, primers based on sequences from the ARTICnetwork were used to generate 400 bp amplicons in two different PCR pools. After merging of pools and amplification, libraries were constructed using QIASeq FX DNA Library UDI Kit following the manufacturer's instructions (Qiagen GmbH, North Rhine-Westphalia, Germany).

Sequencing was done with Illumina NextSeq® 500/550 using 149-bp paired-end reads with 10-bp indices (Illumina, California, USA). Obtained viral sequences were assembled using CLC Genomics Workbench v20.0.3 (Qiagen GmbH, North Rhine-Westphalia, Germany). SARS-CoV-2 isolate Wuhan-Hu-1 served as the reference genome (Accession NC_045512.2). SARS-CoV-2 variants were identified by uploading FASTA files on freely accessible databases (http://cov-lineages.org/).

**Extraction of the buffy coat and purification of RNA**. Whole blood was collected, and total RNA was extracted from the buffy coat and purified using the Maxwell RSC simply RNA Blood Kit (Promega) according to the manufacturer's instructions. The concentration and quality of RNA were assessed by an Agilent Bioanalyzer 2100 (Agilent Technologies, CA).

**Messenger RNA sequencing (mRNA-seq) and data analysis**. The Poly-A containing mRNA was purified by poly-T oligo hybridization from 1 μg of total RNA and cDNA was synthesized using SuperScript III (Invitrogen, MA). Libraries for sequencing were prepared according to the manufacturer's instructions with TruSeq Stranded mRNA Library Prep Kit (Illumina, CA, RS-20020595) and paired-end sequencing was done with a NovaSeq 6000 instrument (Illumina), yielding average of 190 million reads per sample.

The raw data were subjected to QC analyses using the FastQC tool (version 0.11.9) (https://www.bioinformatics.babraham.ac.uk/projects/fastqc/). mRNA-seq read quality control was done using Trimmomatic[20] (version 0.36) and STAR RNA-seq[21] (version STAR 2.5.4a) using 150 bp paired-end mode was used to align the reads (hg19). HTSeq[22] (version 0.9.1) was to retrieve the raw counts. With R (https://www.R-project.org/) languages, DESeq2[23] were used for differential gene analysis and the Wald statistic. Additionally, the RUVSeq[24] package was applied to remove unwanted variation (RUV) (Supplementary Fig. 4). The data were pre-filtered keeping only genes with at least ten reads in total. The visualization was done using dplyr (https://CRAN.R-project.org/package=dplyr) and ggplot2[25]. Genes were categorized as significantly differentially expressed with a *p* value below 0.05 and a fold change > 2 for upregulated genes and a fold change of < −2 for downregulated ones and then conducted gene enrichment analysis (https://www.gsea-msigdb.org/gsea/msigdb).

**Statistical analysis**. RNA-seq data were evaluated with the Wald statistic from DESeq2 and *p* value for all genes were listed in Supplementary Data. Data were presented as the means ± s.e.m. (standard error of the mean) of all experiments with *n* = number of biological replicates. For comparison of RNA expression levels between two groups, data were presented as the standard deviation in each group and were evaluated with a two-way ANOVA followed by Tukey's multiple comparisons test, one-way ANOVA with Dunnett's multiple comparisons, or a two-tailed unpaired *t*-test with Welch's correction using GraphPad PRISM (version

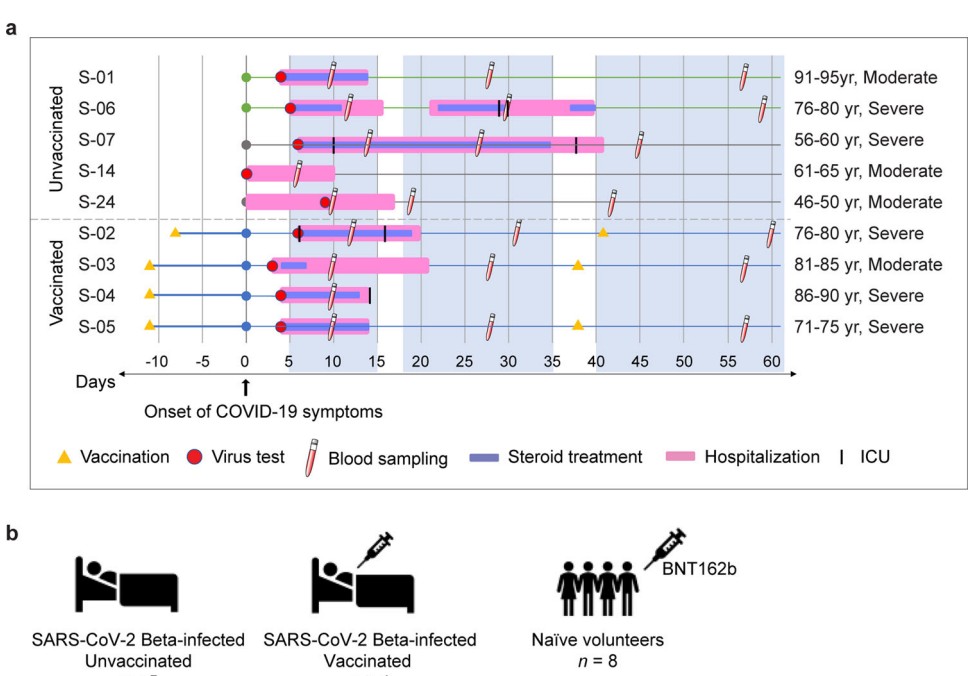

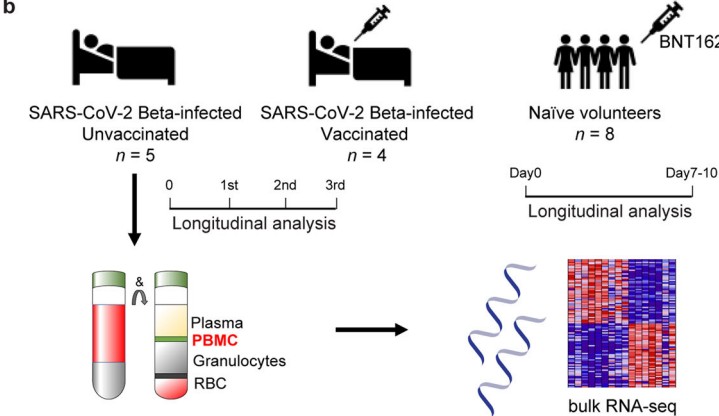

**Fig. 1 Experimental design. a** Overview of the patient cohort. Patients S-01 to S-06 resided in a retirement home in Tyrol and developed COVID-19 at the end of January 2021. Patients S-07 to S-09 were from other parts of Tyrol and South Tyrol. Whole viral genome sequencing was used to validate the identity of the SARS-CoV-2 Beta variant. Age, COVID-19 severity vaccination status, SARS-CoV-2 test, hospitalization duration, treatment duration, and blood sampling time points are shown. For each patient, the day when COVID-19 symptoms were diagnosed was set as 0. **b** Experimental flow. Buffy coat cells were isolated from unvaccinated ($n = 5$) and vaccinated ($n = 4$) hospitalized patients and RNA-seq was conducted. Naïve healthy volunteers were vaccinated with BNT162b2 and RNA-seq was conducted from buffy coats isolated at days 0 and 10.

9.0). A value of *$P < 0.05$, **$P < 0.001$, ***$P < 0.0001$, ****$P < 0.00001$ was considered statistically significant. Power analysis for the data set was performed with all significantly regulated genes, induced genes, their average read counts, and dispersion via RnaSeqSampleSize[26] and Sample Size Calculators for designing clinical research[27]. Our datasets have statistical power (>power 0.8) and the required sample size is four.

**Ethics approval**. This study was approved by the Institutional Review Board (IRB) of the Office of Research Oversight/Regulatory Affairs, Medical University of Innsbruck, Austria (EK Nr: 1064/2021). Written informed consent was obtained from all study participants. The informed consent was checked and approved by the IRB of the Office of Research Oversight/Regulatory Affairs, Medical University of Innsbruck, Austria. The study was carried out in accordance with the Declaration of Helsinki (https://www.wma.net/policies-post/wma-declaration-of-helsinki-ethical-principles-for-medical-research-involving-human-subjects/).

**Reporting summary**. Further information on research design is available in the Nature Research Reporting Summary linked to this article.

## Results

**Study design**. To better understand the immune response of individuals infected with SARS-CoV-2 following their first dose of the BNT162b2 mRNA vaccine, we investigated a cohort of nine hospitalized COVID-19 patients infected with the Beta variant (Fig. 1a). Six individuals (average age 82 years) from a retirement home that experienced an outbreak of the Beta variant in January of 2021 and three Beta-positive patients (average age 56 years) from other communities in Tyrol and South Tyrol (Italy) (Fig. 1a and Supplementary Data 1). Whole viral genome sequencing confirmed the Beta variant. The six patients from the retirement home developed COVID-19 within 2 days of each other, were admitted to the same hospital, treated with Fortecortin (Dexamethasone), and cared for by the same physicians. Three patients from the retirement home had received the first dose of the BNT162b2 (Pfizer-BioNTech) vaccine 11 days prior to the onset of COVID-19 symptoms and one patient 8 days prior. All patients had underlying health conditions and seven received dexamethasone treatment (Supplementary Data 1). Having this well-controlled cohort, provided the opportunity to explore the impact of the first vaccination on the immune transcriptome in hospitalized COVID-19 patients.

**Transient activation of the JAK-STAT pathway in hospitalized patients infected after the first vaccination**. Blood was drawn

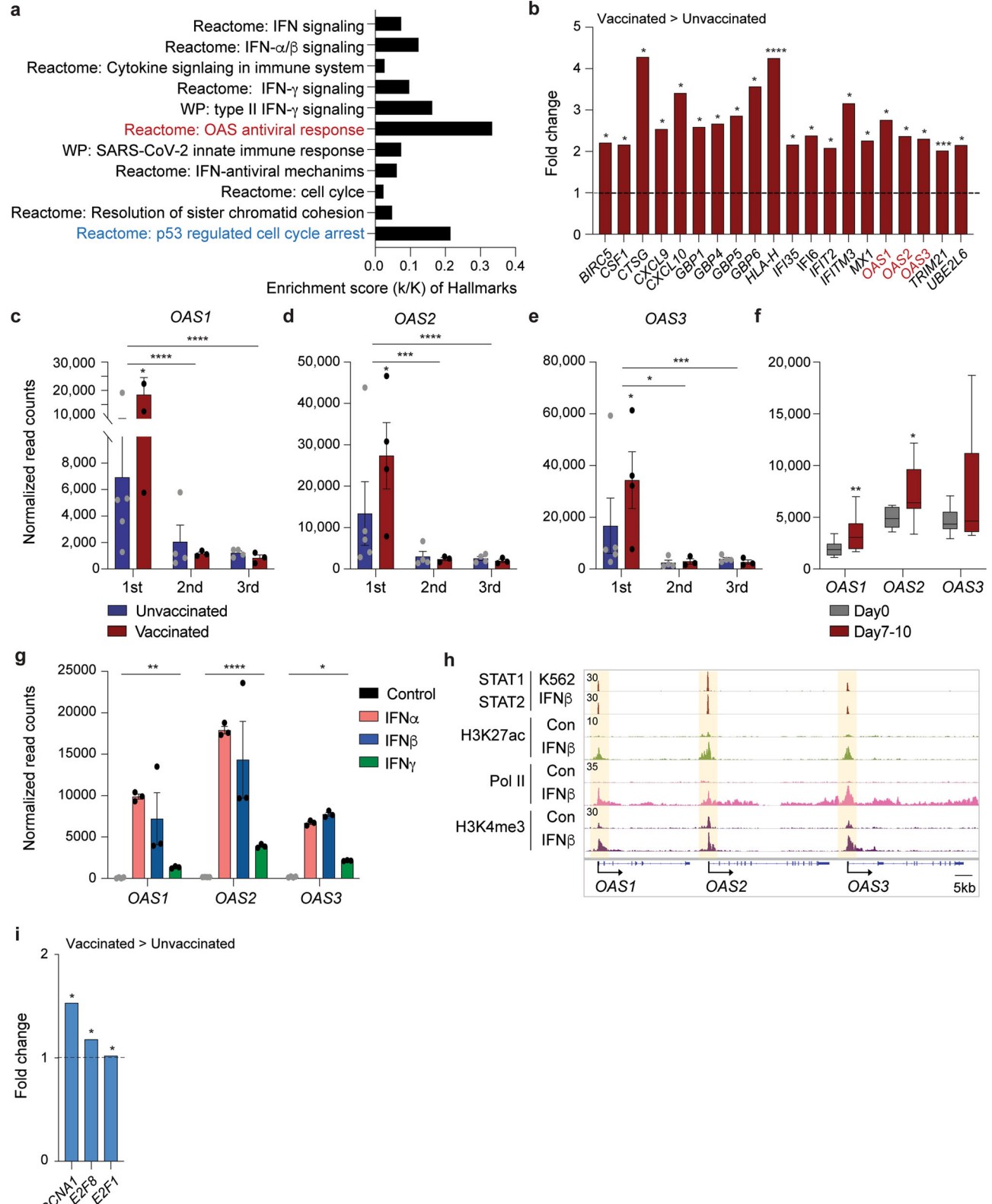

during three distinct time windows after the development of COVID-19 symptoms, RNA was prepared from buffy coats followed by RNA-seq analyses (Fig. 1). First, we analyzed the transcriptomes at approximately day 10 after the first symptomology (Supplementary Data 2). A total of 522 genes were significantly and at least twofold differentially expressed between the four vaccinated patients and the five unvaccinated ones. The power analysis showed significance for these datasets (Methods section). The expression of 166 genes was elevated at least twofold in the vaccinated group compared to the unvaccinated one (Supplementary Data 2). The significantly upregulated genes were enriched in immune response pathways, including interferon-JAK/STAT signaling, influenza vaccination[28], and OAS antiviral response (Fig. 2a, b and Supplementary Data 2).

**Fig. 2 Genetic programs preferentially activated in vaccinated COVID-19 hospitalized patients compared to unvaccinated ones. a** Canonical pathway analysis of significantly enriched genes in clusters for innate immune-related biological processes, including OAS antiviral response. **b** Relative mRNA levels of 20 genes that are induced in vaccinated patients as compared to unvaccinated ones and enriched in IFN signaling are presented by bar graphs. A *p* value by the Wald statistic from DESeq2 were listed in Supplementary Data 2. *$p < 0.05$, ***$p < 0.0001$, ****$p < 0.00001$. **c–f** Relative normalized expression of OASs gene from unvaccinated and vaccinated patients in longitudinal analysis (**c–e**) as well as Day 0 and Days 7–10 of naïve volunteers receiving the first dose of the BNT162b2 vaccine (**f**). A two-tailed unpaired *t*-test with Welch's correction was used to evaluate the statistical significance between two groups as well as two time points. *$p < 0.05$, **$p < 0.001$, ***$p < 0.0001$, ****$p < 0.00001$. **g** OAS1-3 mRNA levels from primary bronchial cells untreated or treated with different IFNs were measured by RNA-seq. Results are shown as the means ± s.e.m. of independent biological replicates ($n = 3$). Two-way ANOVA followed by Tukey's multiple comparisons test was used to evaluate the statistical significance of differences. **h** ChIP-seq for STAT transcription factors, the histone marks H3K27ac and H3K4me3, and Pol II was conducted in primary bronchial cells at the absence and presence of IFNβ[12]. **i** Relative mRNA levels of three genes that are induced in vaccinated patients as compared to unvaccinated ones and enriched in p53 regulated cell cycle arrest are presented by bar graphs. A *p* value by the Wald statistic from DESeq2 were listed in Supplementary Data 2. *$p < 0.05$.

Specifically intriguing was the identification of pathways and defined genes linked to antiviral response[29] that were distinctly induced in the immune transcriptome of vaccinated hospitalized patients. The OAS antiviral response attracted our attention as mutations (haplotypes) protective from severe COVID-19 had been identified in OAS1[30] and traced to the Neanderthal genome[15,31]. The OAS locus encodes three antiviral 2′,5′-oligoadenylate synthetase (OAS) enzymes, which sense and bind dsRNA, leading to the activation of the RNase L pathway and the degradation of viral RNA, inhibition of virus replication, and cell death[32]. While innate immune response programs were induced in unvaccinated COVID-19 patients within the first 10 days after symptomology (Supplementary Data 2), the expression of the three OAS genes was further elevated in the vaccinated group (Fig. 2c–e and Supplementary Fig. 1). The induction of the OAS1 gene was transient, and levels decreased by ~94% at day 35 post symptomology (Fig. 2c and Supplementary Data 3, 4).

The preferential induction of the three OAS genes in vaccinated as compared to unvaccinated hospitalized patients opened the possibility that vaccination by itself could result in increased OAS expression. To test this possibility, we conducted a separate study on eight healthy naïve individuals receiving their first dose of the BNT162b2 vaccine. RNA-seq was conducted on Buffy coats harvested just prior to the vaccination (Day 0) and 7 to 10 days later (Day 7–10) (Supplementary Data 5). A significant increase of OAS1 and OAS2, but not OAS3, mRNA levels was observed upon vaccination (Fig. 2f). Since the induction of OAS expression in the vaccinated patients exceeded that seen in the vaccinated naïve individuals it is likely that SARS-CoV-2 infection and the development of COVID-19 triggered a booster.

Although the OAS genes are known as interferon-stimulated genes (ISG), there is limited information on their regulation through the JAK-STAT pathway. Next, we investigated the impact of individual interferons in OAS regulation (Fig. 2g, h). We recently conducted a genome-scale study on primary bronchial cells treated with different interferons[12] and analysis of these data demonstrated that IFNα and IFNβ induced OAS1 and OAS2 expression ~100-fold (Fig. 2g). In contrast, induction of OAS3 was less than 50-fold. IFNγ had less impact on the activation of the OAS locus. To obtain structural insight into regulatory elements controlling the OAS locus, we analyzed ChIP-seq experiments from primary bronchial cells induced with IFNβ[12]. IFNβ induced H3K27ac and H3K4me3 coverage at the promoter/upstream regulatory region as well as Pol II loading (Fig. 2h). ChIP-seq data from K562 cells demonstrated STAT1/2 binding to promoter sequences, pointing to a key role of these transcription factors. Mutations in the OAS1 gene have been identified and one linked to the Neanderthal genome appears to protect from COVID-19 susceptibility and severity[15]. Three of the nine COVID-19 patients were homozygous for the

Neanderthal mutation (Supplementary Data 6) and no correlation could be made with the disease severity.

In coronavirus infection, similar to other viruses, p53 has been shown to play a role in limiting viral replication.[33,34] p53 degradation triggered by coronavirus Nsp3 protein is suggested to contribute to virus survival. We found that vaccinated patients demonstrated enrichment in p53 regulated cell cycle arrest genes, specifically CCNA1, E2F8, and E2F1 (Fig. 2a, i), consistent with a possibility of higher p53 activity levels in the vaccinated patients. Interestingly E2F8, but not CCNA1 or E2F1, was modestly upregulated 7 days post-vaccination in the naïve vaccinated individuals (1.5-fold, $p_{adj} = 0.04$).

DHX58 (LGP2), originally identified in the mammary gland[35], is a member of the family of retinoic acid-inducible gene I (RIG-I)-like receptors (RLRs), which are key sensors of virus infection[36,37]. Collectively, the three family members mediate the transcriptional induction of type-I interferons and antiviral host response genes. Expression of DHX58 increased more than two-fold in the vaccinated hospitalized cohort (Fig. 3a). In contrast, the expression of the other two members IFIH1 (MDA-5) and DDX58 (RIG-I) was not significantly induced (Fig. 3b). Primary cell culture experiments demonstrated that IFNα and IFNβ induced all three RLR genes (Fig. 3c, d). ChIP-seq analyses revealed the location of putative regulatory elements and identified STAT1/2 as a potential key regulator of these genes (Fig. 3e). Lastly, we investigated the regulation of the genes encoding Interferon Induced proteins with Tetratricopeptide repeats (IFIT) proteins, which confer immunity against viral infection. From the 11 members, only IFITM3, linked to COVID-19[38,39], was induced in vaccinated patients (Fig. 3f).

**Longitudinal analysis of vaccinated and unvaccinated hospitalized patients.** From our cohort of the four vaccinated patients, one died of COVID-19 and three were discharged from the hospital after a single stay of 10–14 days (Fig. 1a). Of the five non-vaccinated patients, four were discharged after a single stay and one was re-admitted. To dig deeper into the temporal progression of the immune response after discharge from the hospital, we analyzed the immune transcriptomes of the three vaccinated patients ~3 and 6 weeks after the first transcriptome analysis (Figs. 1a, 4a and Supplementary Data 3). Similarly, we investigated the immune transcriptome of the four unvaccinated patients after their discharge from the hospital (Figs. 1a, 4b and Supplementary Data 4). Gene enrichment analyses of differentially expressed transcripts illustrate a greatly diminished immune response, including INFα/γ signaling, in the patients recovered from COVID-19 (Fig. 4c, d and Supplementary Data 3, 4). In general, expression of programs linked to innate immunity, including specific genes, described earlier, were approaching levels seen in non-infected naïve individuals (Supplementary

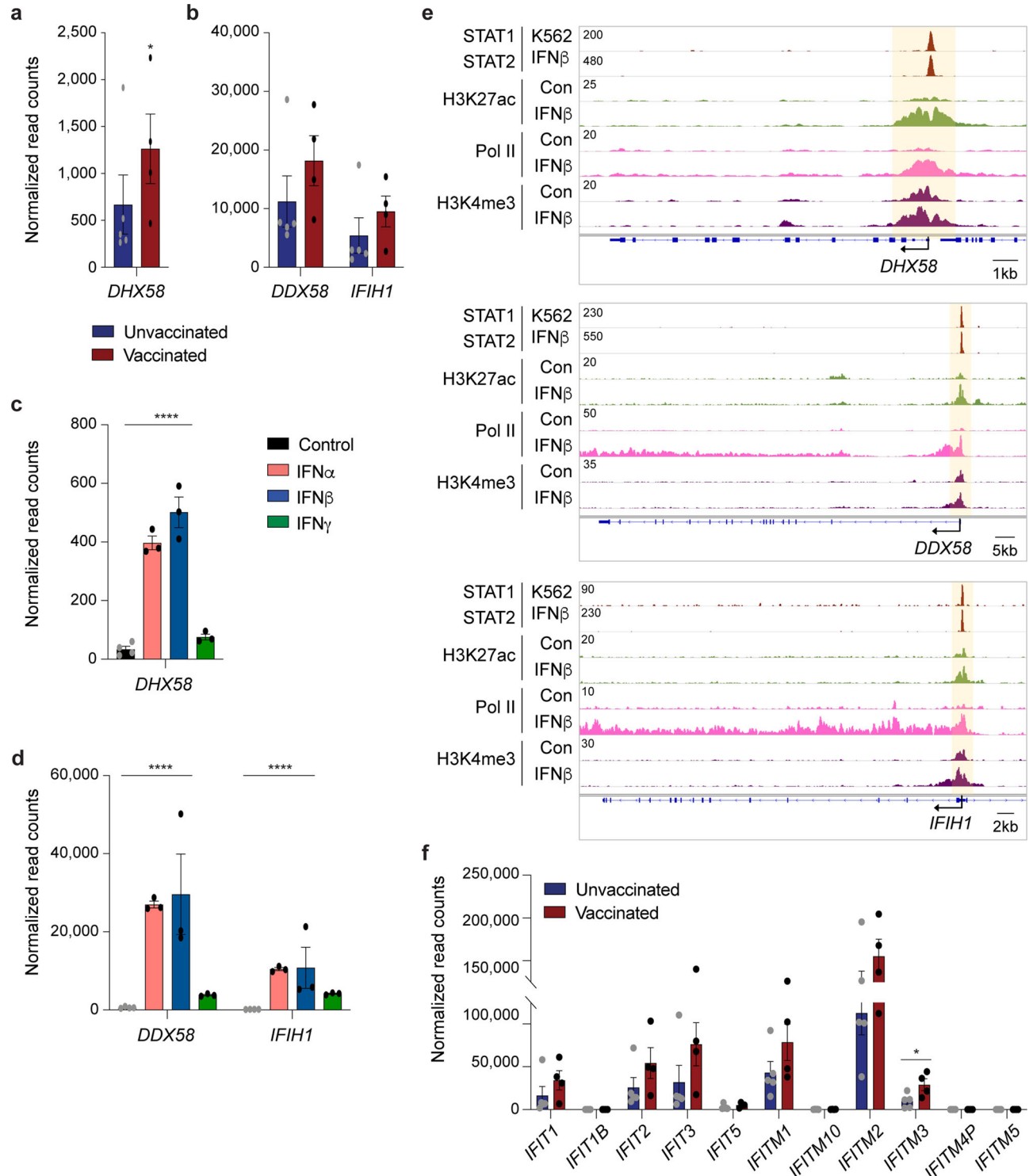

**Fig. 3 Regulation of innate immune response genes. a**, **b** Relative normalized expression of the RIG-I-like receptors family members *DHX58* (**a**), *DDX58* and *IFIH1* (**b**) in vaccinated and unvaccinated patients. A two-tailed unpaired *t*-test with Welch's correction was used to evaluate the statistical significance between the two groups. *$p < 0.05$ **c**, **d** *DHX58* (**c**), *DDX58* and *IFIH1* (**d**) mRNA levels from primary bronchial cells treated with IFNs were measured by RNA-seq[12]. Results are shown as the means ± s.e.m. of independent biological replicates ($n = 3$). ****$p < 0.00001$ **e** ChIP-seq for STAT transcription factors, the histone marks H3K27ac and H3K4me3, and Pol II was conducted in primary bronchial cells in the absence and presence of IFNβ. **f** mRNA levels of IFIT gene family members in vaccinated and unvaccinated patients. A two-tailed unpaired *t*-test with Welch's correction was used to evaluate the statistical significance between the two groups.

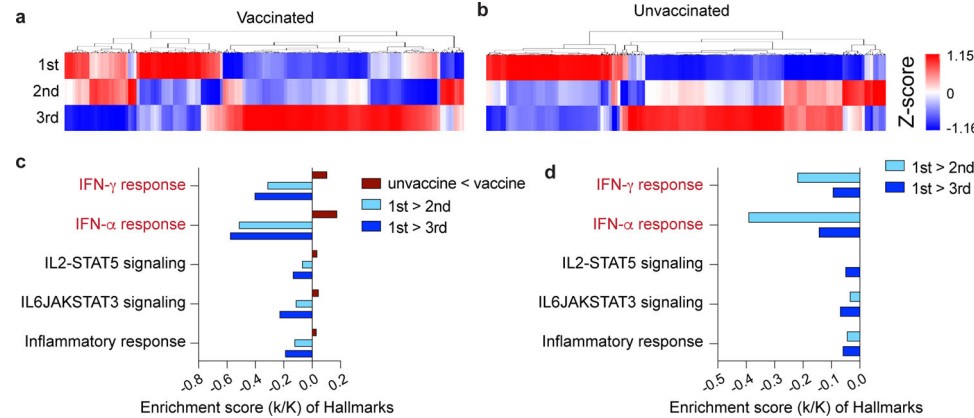

**Fig. 4 Longitudinal transcriptome study. a**, **b** Heatmap shows the expression levels of 522 genes significantly regulated in vaccinated (**a**) and unvaccinated (**b**) patients throughout the 6-week longitudinal study (analysis of samples from the three time windows). Each sample is hierarchically clustered by using their adjacency scores as distance. **c**, **d** Genes reduced significantly in the longitudinal analysis of vaccinated (**c**) and unvaccinated (**d**) were significantly enriched in Hallmark Gene Sets (FDR $q$ value < 0.005). All hallmarks are related to immune regulation.

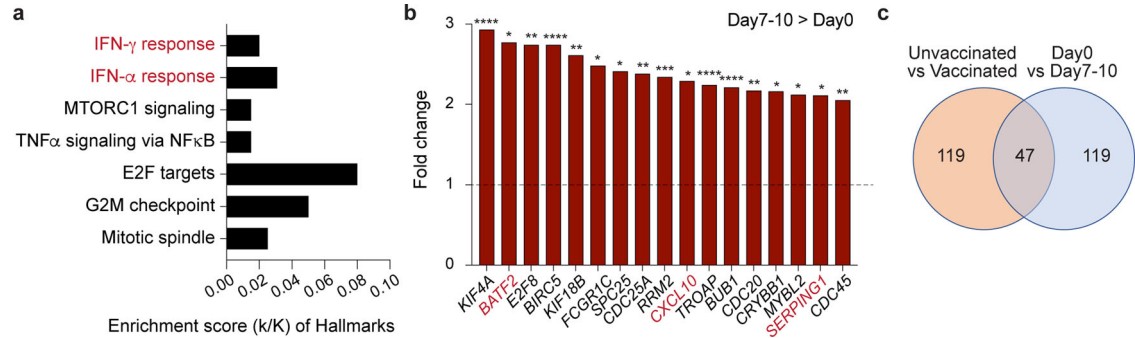

**Fig. 5 Immune transcriptome induced upon BNT162b2 vaccination. a** Genes expressed at significantly higher levels at Day 7–10 from the vaccination were significantly enriched in Hallmark Gene Sets (FDR $q$ value < 0.005). **b** Relative mRNA levels of 17 genes that are induced at Day 7–10 as compared to Day 0 and enriched in IFN signaling are presented by bar graphs. A $p$ value by the Wald statistic from DESeq2 were listed in Supplementary Data 5. *$p$ < 0.05, ***$p$ < 0.0001, ****$p$ < 0.00001. **c** Transcriptional signatures in vaccinated naïve individuals and vaccinated COVID-19 patients. The Venn diagram displays the number of significantly induced genes.

Data 7, 8). However, adaptive immunity was also detected in the vaccinated patients compared to non-COVID samples.

**Immune transcriptomes induced by BNT162b2 vaccination of healthy naïve volunteers.** Systems vaccinology[16] addresses immune responses to vaccines through genome-scale analyses[17], including RNA-seq[18]. Next, we addressed the question to what extent the 166 gene set preferentially induced in vaccinated hospitalized patients was also induced by the vaccination of healthy naïve individuals in the absence of COVID-19. We conducted a BNT162b2 vaccination study on eight healthy naïve volunteers (age 27–51 years). RNA-seq was conducted on buffy coats harvested just prior to the vaccination (Day 0) and after 7–10 days, with the data set having a power of 0.95. We identified 79 genes that were induced at least two-fold after the first dose of BNT162b2 (Supplementary Data 5) and they are enriched in IFNα/γ responses (Fig. 5a). Seventeen of these genes coincided with the 166 gene set identified in the vaccinated patient population (Fig. 5b) and three genes are components of IFN signaling pathways (marked in red). An additional 30 genes, including *OAS* and *DHX58*, were significantly induced in the vaccinated naïve cohort that matched the 166 gene set (Fig. 5c and Supplementary Data 5). Notably, key genes involved in innate sensing of viruses, such as *OAS* family members and *DHX58* were also identified

upon yellow fever vaccination[17] suggesting that they are part of a broad vaccine-induced innate immune response. The *ACE2* gene, encoding the receptor of SARS-CoV-2, is regulated by interferons through the JAK-STAT pathway[12,40] opening the possibility of increased expression in hospitalized COVID-19 patients. Notably, *ACE2* mRNA levels were significantly elevated in hospitalized patients regardless of their vaccination status as compared to healthy volunteers receiving the first dose of BNT162b2 (Supplementary Fig. 2). The *ACE2* gene has two distinct promoters, one controlling the full-length mRNA and an intronic one responsible for the short form of ACE2[12,41–43]. SARS-CoV-2 can infect immune cells[44] and an increase of the native ACE2 would elevate the propensity of infection. The number of overlapping genes significantly regulated between the gene expression of naïve vaccinated individuals and vaccinated/unvaccinated COVID-19 patients is summarized in Supplementary Fig. 3. Expanding RNA-seq studies to other vaccines should identify not only a core "vaccine signature" but possibly also vaccine-specific response.

**Discussion**
Our study shed light on the immune transcriptome that was preferentially activated in hospitalized COVID-19 patients that had received the first dose of the mRNA vaccine BNT162b2 prior to developing the disease, as compared to unvaccinated patients.

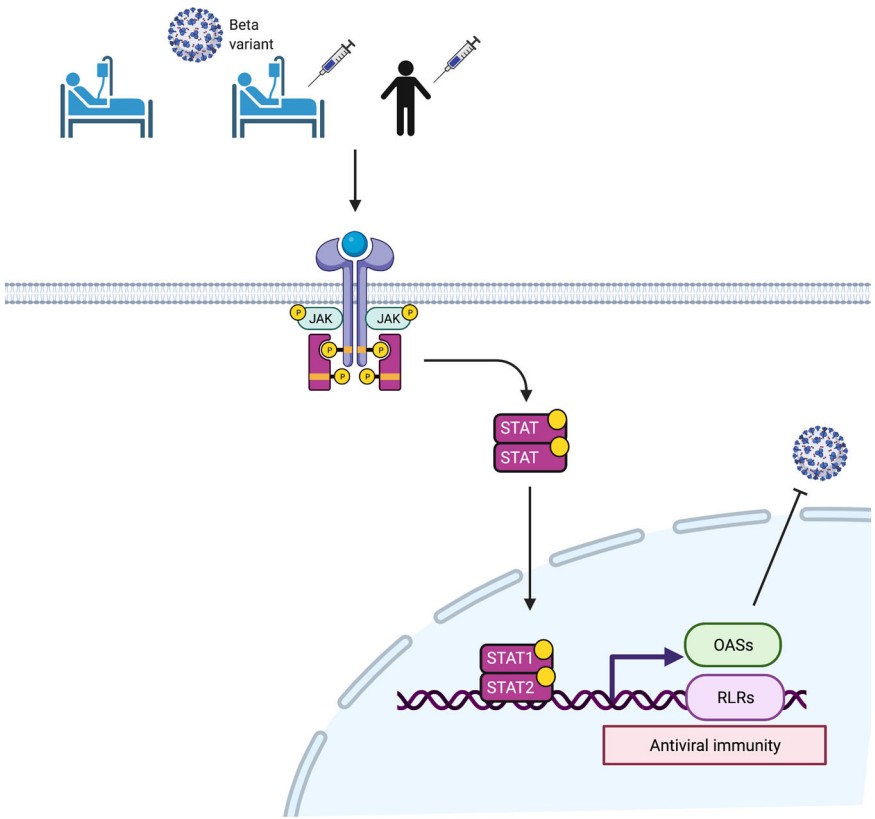

**Fig. 6 Illustration of JAK-STAT-dependent antiviral immunity by vaccination.** BNT162b2 vaccination activates the JAK-STAT pathway and active STAT transcription factors induce the expression of *OAS*s and RIG-I-like receptors (RLR) family members related to antiviral response. This figure was created with BioRender.com (Agreement number: ZJ23HFYMY6).

In addition to the well-established dysregulated innate immune responses observed in COVID-19 patients[13,45], we observed a hyperregulation of a 166 gene set, with key antiviral genes under the control of the JAK-STAT signaling pathway. STAT1 has been shown to regulate immune pathologies in a mouse model of SARS-CoV infection[46]. A subset of 47 genes was also induced, although at a lower level, in naïve healthy volunteers after receiving their first dose of BNT162b2. These findings suggest that SARS-CoV-2 infection coupled with severe COVID-19 boosts the induction of genetic programs primed by the BNT162b2 vaccine (Fig. 6). Collectively, our studies provide insight into immune responses induced by mRNA vaccination in naïve individuals and in patients subsequently infected with SARS-CoV-2 infection.

The intersection of vaccination and COVID-19 is a rapidly evolving field in which real-world data, especially those concerning variants of concern (VOC) and the impact of prior vaccination, is critical for everyday management of the COVID-19 disease by public health officials, government agencies, and institutions. It is known that while the first vaccine dose is not fully protective[47], data available indicates that as early as 12 days following the first vaccination there may be a response against the SARS-CoV-2 Alpha and Beta variants[6,48]. Our work documented a stronger molecular immune response in patients that had received their first dose of BNT162b2 prior to infection with the Beta variant but no apparent clinical benefit. We show that for elderly patients, reflective of groups with a higher inherent risk of morbidity and mortality from SARS-CoV-2 infection, the first vaccine provided minimal clinically significant protection if any. We did not observe a difference in disease severity between the vaccinated and unvaccinated groups and one of the vaccinated individuals died.

Our study illustrates that whole transcriptome investigation from the buffy coat is feasible and yields actionable data when performed in well-controlled clinical settings. It demonstrates a technique that is feasible to expand and reproduce across settings. While our study was limited to infection ~11 days following the first vaccination, extensions could compare the immune transcriptome after the second vaccination and investigate individuals who may have waning antibody response ten or more months following either infection and/or full immunization. As a caveat, it can be challenging to reliably compare RNA-seq data generated on different platforms and the disease severity at the time of the analysis greatly impacts the transcriptome[13].

The findings in this report are subject to at least three limitations. The transcriptomes from patients infected with the Beta variant but not with other variants have been investigated. This study focused on elderly patients who are particularly vulnerable to COVID-19. The effects of the BNT162b2 mRNA vaccine have been investigated but not of any other vaccine.

Further research in systems vaccinology will continue to address mechanisms to improve the protective immunological response to SARS-CoV-2 through boosters and vaccines engineered specifically for variants. Serial immune transcriptome studies should be included in addition to antibody tests for a fuller understanding of the spectrum of immune response in real-world situations as more immunized individuals encounter breakthrough infections. Such studies should also contribute to an understanding of how best to manage the timing of third boosters, variant targeting, and addressing the vaccinations of immune-compromised individuals.

## Data availability

Complete SARS-CoV-2 genome sequences have been deposited in GISAID with accession numbers EPI_ISL_9139218 to EPI_ISL_9139224 (https://www.gisaid.org/). The RNA-seq datasets generated during the current study are available in the GEO repository with accession number GSE189039. The RNA-seq and ChIP-seq datasets from primary bronchial cells[12] analyzed during the current study are available in the GEO repository under accession number GSE161665. The ChIP-seq data for STAT1[49] are available in the GEO repository under accession number GSE31477. All other data are available from the corresponding author on reasonable request.

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

## Acknowledgements

Our gratitude goes to the patients who contributed to this study to advance our understanding of COVID-19. This work was supported by the Intramural Research Programs (IRPs) of the National Institute of Diabetes and Digestive and Kidney Diseases (NIDDK) and utilized the computational resources of the NIH HPC Biowulf cluster (http://hpc.nih.gov). RNA sequencing was conducted in the NIH Intramural Sequencing Center, NISC (https://www.nisc.nih.gov/contact.htm).

## Author contributions

L.K.: recruited patients, collected material, and analyzed data; H.K.L., P.A.F., and L.H.: analyzed data and wrote the manuscript; M.W.: collected and prepared material; A.M., A.Z., L.K.Sr., S.R., M.B., J.S., and N.K.: recruited and diagnosed patients. All authors read and approved the manuscript.

## Funding

## Competing interests

The authors declare no competing interests.
