## [Peer Review File · Communications Medicine]

Reviewers' comments:

Reviewer #1 (Remarks to the Author):

In this study by Knabl, et al., the investigators use RNA-Seq to investigate the transcriptome of 6 elderly individuals ill, hospitalized with B.1.351 SARS-CoV-2 infection, four of whom had received at the first dose of the Pfizer BNT162b vaccine 8-11 days prior to the onset of COVID-19 symptoms. All patients were moderately to severely ill, and 1 patient (in the partially vaccinated group) died. They look at the transcriptomes at days 10 and 35 following onset of COVID-19 symptoms. Compared to unvaccinated individuals (n=2), vaccinated patients (n=4) had a more intense immune response, with relative upregulation of JAK/STAT pathway, interferon-associated, and innate viral immunity genes. They found that the transcriptomes from vaccinated individuals with COVID-19 were more similar to younger individuals with COVID-19 than unvaccinated older individuals with COVID-19. As previously shown, partial vaccination did not appear to be sufficient in protecting against severe disease.

The strengths of this study are the availability of longitudinal samples from the same patients and also the new findings gleaned from transcriptome analysis of infected vaccinated individuals. In general, the topic is of great interest to SARS-CoV-2 researchers and more broadly with respect to host response in vaccination and breakthrough infections. This study has a number of weaknesses, though, some of which are addressable by additional experiments or more details to provide clarity, but others which are due to the fundamentally limited nature of the study design and analyses, as follows:

(1) A key limitation of the study is the lack of a comparison group of individuals who were vaccinated but uninfected. Thus, it is very challenging to separate the contribution to the immune/inflammatory response from vaccination alone versus SARS-CoV-2 alone. Did vaccination “prime” the immune system such that there is an exaggerated inflammatory response to subsequent infection? Or did vaccination itself result in a baseline level of inflammation?

(2) The gene expression profile obtained by RNA-Seq is very sensitive to the choice of controls used. The manuscript just references another paper from the authors but no further details on the controls used for the current study are given. Are the controls healthy matched donors, “highly exposed [SARS-CoV-2] seronegative individuals”, or some other group? Were the controls symptomatic with respiratory disease or asymptomatic or did they have an inflammatory/infectious condition other than SARS-CoV-2? A detailed description of the controls used for this study, the number of controls, how the controls were selected, etc. is needed.

(3) The number of patients looked at is very small (4 vaccinated, 2 unvaccinated). I am skeptical regarding the validity of any conclusions made regarding the transcriptomes from such a small dataset. A larger “n” is required, especially to make conclusions such as that the transcriptomes of vaccinated individuals are similar to younger individuals with COVID-19 versus unvaccinated older individuals. The finding that partial vaccination (8-11 days after the first dose) is not protective against severe disease is already well established.

(4) The PCA plots in Figures 1 and 2 are confusing. PCA plots are multi-dimensional. Why are they only clustering according to the one axis (PC1)? Why is there no evidence of clustering with the PC2 and the other undisplayed axes (PC3, PC4, etc.)? Were the PCA plots generated from all ~23,000 genes in the transcriptome or only a subset of genes, and if the latter, how were the subset of genes

selected?

(5) In Figures 1c and 2c, what is the distribution of gene categories in the gray area (genes that are not virus infection related)?

(6) In Figure 2b, how were the specific differentially expressed genes selected?

(7) The expression profile is likely to be biased by the severity of the disease in these patients (all were hospitalized). As an example, the difference in ACE counts looks almost too good to be true. The distribution contrasts greatly with that from much larger studies of the transcriptome in SARS-CoV-2 patients (for example, please see Figure 4b in Butler, et al. 2021, Nature Communications, 12:1660, <https://www.nature.com/articles/s41467-021-21361-7>).

(8) There are typos and unusual grammatical constructions throughout the manuscript. "old people's home" = "senior home", "firsst" should be "first" in Figure 1, "Immune transcriptomes were established" should be "Immune transcriptomes were obtained", "documented infection" should be "documented", etc. Please review the manuscript carefully to correct these.

Reviewer #2 (Remarks to the Author):

The study "Impact of BNT162b first vaccination on the immune transcriptome of elderly patients infected with the B.1.351 SARS-CoV-2 variant" by Knabl L et al. analyzed PBMC transcriptomes from a unique cohort of the elderly, either vaccinated or not, who was infected by SARS-CoV-2 variant B.1.351. The authors analyzed differential gene expression for pathway enrichment analysis and concluded that "the transcriptomes of the older vaccinated group significantly different than older unvaccinated individuals infected at the same institution and more similar to the immune response of younger unvaccinated individuals (ages 48-62) following B.1.351 infection." Although the study on this unique cohort is potentially interesting and might be to provide some insights into how vaccine influences immune response, the bioinformatic analysis was very basic and, sometimes, was not conducted in a rigorous manner.

Questions

1) There was no sufficient information for data analysis. In methods, "R (<https://www.R-project.org/>), Bioconductor and DESeq2 were used." Such a description is very generic. Please detail how each package was applied in what particular analysis. "the RUVSeq32 package was applied to remove confounding factors." Please describe how the batch effect was corrected. In Fig. 1b, samples from three groups were very separated in PC1 but largely overlapped in PC2. I was concerned with the potential batch effects. Could the authors detail whether libraries or sequencing were conducted in one batch or different batches?

2) There is no sufficient information for human subjects. It is essential to have the information of human subjects as a part of the methods. What were the underlying conditions for each patient? Were the doses of dexamethasone therapy in all patients the same or different? Besides dexamethasone, were there other therapies that potentially affect immune response? It seems that non-COVID controls were from a different study. The nature of these controls should also be included.

3) The GSEA was a good starting point but further analyses are required to understand how individual pathways were enhanced or suppressed. It was unclear how virus infection related genes (Fig. 1c) or vaccine related immune genes (Fig. 2c) were defined?

4) In Fig 4, there was a separation between the elderly and younger samples. I don't know why such analysis supports the conclusion in the abstract "the transcriptomes of the older vaccinated group ... more similar to the immune response of younger unvaccinated individuals".

5) There are much inaccurate and missing information throughout the manuscript.

For example.

Line 74 "Four patients had received the first dose of the BNT162b (Pfizer-BioNTech) vaccine 11 days before the onset of COVID-19 symptoms and one patient eight days prior (Fig. 1a)." It should be "Three... and one"

Fig. 1b, 2a, 4a should have labelled subject ID

Fig 2e, X-axis is not labelled.

Fig 4a, Better labelled as old vs middle-aged, 48-62 yr old are not "young"

Response to the reviewers

We thank the reviewers for evaluating our work and suggesting a path forward.

While our manuscript was under review we conducted additional studies, which are now included in the revised manuscript. Major additions to the manuscript include:

1. Immune transcriptome study on naïve healthy individuals receiving their first dose of the BNT162b vaccine.
2. Additional longitudinal RNA-seq data from our hospitalized patients.
3. Combining the group of unvaccinated ($n=5$) and vaccinated patients ($n=4$) and conducting a power analysis.
4. Identification of genes controlling viral infection (*OAS1*, *DHX58* and others) and linked to COVID-19 disease.
5. Identification of the 'Neanderthal' haplotype in our hospitalized COVID-19 patients.
6. Additional experiments investigating the regulation of key JAK-STAT pathway genes identified in our study. Analyses of RNA-seq and ChIP-seq data from primary cells.
7. Integration of the immune transcriptome data from the vaccinated hospitalized patients and the vaccinated healthy naïve individuals.

Reviewer #1

As the reviewer pointed out, *“the strengths of this study are the availability of longitudinal samples from the same patients and also the new findings gleaned from transcriptome analysis of infected vaccinated individuals. In general, the topic is of great interest to SARS-CoV-2 researchers and more broadly with respect to host response in vaccination and breakthrough infections”*.

(1) A key limitation of the study is the lack of a comparison group of individuals who were vaccinated but uninfected. Thus, it is very challenging to separate the contribution to the immune/inflammatory response from vaccination alone versus SARS-CoV-2 alone. Did vaccination “prime” the immune system such that there is an exaggerated inflammatory response to subsequent infection? Or did vaccination itself result in a baseline level of inflammation?

Response

We fully agree with the reviewer. While our manuscript was under review, we have conducted a study on healthy naïve volunteers ($n=8$) receiving their first dose of BNT162b and these data are now included in the manuscript (Figure 5 and Supplementary Tables 5, 7 and 8). We used RNA-seq to generate immune transcriptomes from PBMCs at day 0 (just prior to the vaccination) and at day post vaccination. The day 0 group serves as a control. These experiments have demonstrated that 47 genes out of 166 elevated in vaccinated hospitalized patients are

induced by the vaccine in naïve non-COVID individuals. However, their induction in patients is significantly higher than in the vaccine only group, suggesting that SARS-CoV-2 infection or the development of COVID-19 serves as booster.

(2) The gene expression profile obtained by RNA-Seq is very sensitive to the choice of controls used. The manuscript just references another paper from the authors but no further details on the controls used for the current study are given. Are the controls healthy matched donors, “highly exposed [SARS-CoV-2] seronegative individuals”, or some other group? Were the controls symptomatic with respiratory disease or asymptomatic or did they have an inflammatory/infectious condition other than SARS-CoV-2? A detailed description of the controls used for this study, the number of controls, how the controls were selected, etc. is needed.

Response

In the original analysis we had used healthy individuals from our previous study (PMID: 33608566) as the controls. In the revision, we compared the group of five unvaccinated and four vaccinated patients, all of which had been infected with the Beta variant. We also compared four vaccinated Beta-infected patients with eight vaccinated healthy individuals that had enrolled in our vaccination study.

(3) The number of patients looked at is very small (4 vaccinated, 2 unvaccinated). I am skeptical regarding the validity of any conclusions made regarding the transcriptomes from such a small dataset. A larger “n” is required, especially to make conclusions such as that the transcriptomes of vaccinated individuals are similar to younger individuals with COVID-19 versus unvaccinated older individuals. The finding that partial vaccination (8-11 days after the first dose) is not protective against severe disease is already well established.

Response

In the original manuscript we had treated the unvaccinated group ($n=2$) from the retirement home separately from the unvaccinated group ($n=3$) from other hospitals. We have now combined these two groups and have five unvaccinated hospitalized patients and four vaccinated hospitalized patients. The power was adequate to identify differentially expressed genes. We used two programs, RnaSeqSampleSize (PMID: 29843589) and Sample Size Calculators for designing clinical research (Designing clinical research: an epidemiologic approach. 4th ed) to estimate samples size with power 0.8 from the values and significant of differential expressed genes in our data set. The estimated samples size is four per group. Therefore, we analyzed a sufficient number of samples in each group. This is now detailed in the Methods section.

We also included data from additional timepoints up to 55 days post symptomology. This extended longitudinal study also permitted us to investigate the response in individual patients, rather than combing them in cohorts (Supplementary Figure 1).

(4) The PCA plots in Figures 1 and 2 are confusing. PCA plots are multi-dimensional. Why are they only clustering according to the one axis (PC1)? Why is there no evidence of clustering with the PC2 and the other undisplayed axes (PC3, PC4, etc.)? Were the PCA plots generated from all ~23,000 genes in the transcriptome or only a subset of genes, and if the latter, how were the subset of genes selected?

Response

We deleted the PCA plots as our work now focuses more on the JAK-STAT pathway and individual gene classes induced by vaccination of naïve and healthy individuals and in vaccinated COVID-19 patients.

(5) In Figures 1c and 2c, what is the distribution of gene categories in the gray area (genes that are not virus infection related)?

Response

After refocusing our manuscript and including new data, this figure was not needed any longer.

(6) In Figure 2b, how were the specific differentially expressed genes selected?

Response

Genes were categorized as significant differentially expressed with an p-value below 0.05 and a fold change > 2 for up-and down-regulated genes and a fold change of < -2 for down-regulated ones and then conducted gene enrichment analysis (GSEA). This had been described in the Method section.

(7) The expression profile is likely to be biased by the severity of the disease in these patients (all were hospitalized). As an example, the difference in ACE counts looks almost too good to be true. The distribution contrasts greatly with that from much larger studies of the transcriptome in SARS-CoV-2 patients (for example, please see Figure 4b in Butler, et al. 2021, Nature Communications, 12:1660, <https://www.nature.com/articles/s41467-021-21361-7>).

Response

We agree that the expression profiles of individual hospitalized patients reflect the degree of disease severity, the stage of COVID-19, underlying health conditions and the vaccine status, just to name a few variables. Having said this, these are the data from a real-world study with all its pros and cons. Clearly, there are less variables in the group of healthy naïve individuals receiving their first dose of BNT162b. ACE2 expression is equivalent in both patient populations. We thank the reviewer for pointing out the *Nat Comm* paper, which we included into our citations.

(8) There are typos and unusual grammatical constructions throughout the manuscript. “old people’s home” = “senior home”, “firsst” should be “first” in Figure 1, “Immune transcriptomes were established” should be “Immune transcriptomes were obtained”, “documented infection” should be “documented”, etc. Please review the manuscript carefully to correct these.

Response

We carefully edited the manuscript to avoid typos and grave grammatical errors.

Reviewer #2:

Although the study on this unique cohort is potentially interesting and might be to provide some insights into how vaccine influences immune response, the bioinformatic analysis was very basic and, sometimes, was not conducted in a rigorous manner.

Questions

1) There was no sufficient information for data analysis. In methods, “R (<https://www.R-project.org/>), Bioconductor and DESeq2 were used.” Such a description is very generic. Please detail how each package was applied in what particular analysis. ” the RUVSeq32 package was applied to remove confounding factors.” Please describe how the batch effect was corrected. In Fig. 1b, samples from three groups were very separated in PC1 but largely overlapped in PC2. I was concerned with the potential batch effects. Could the authors detail whether libraries or sequencing were conducted in one batch or different batches?

Response

The programs are used to normalize the number of raw RNA reads and analyze differential gene expression (DESeq2). We added information about what each program worked in the analysis in the Methods.

We generated PCA plots with all genes in RNA-seq data. As the reviewer stated, PCs are shown by how much they describe the data. PC1 reveals the most variation, while PC2 reveals the second most variation. For these reasons, differences among clusters along PC1 axis are actually larger than the similar-looking distances along PC2 axis. As each patient has different underlying disease and were hospitalized at that time, the overlapped PC2 level could be from the reasons when the patients were compared to non-COVID controls.

In the revision, we deleted the PCA plots as our work now focuses more on the JAK-STAT pathway and individual gene classes induced by vaccination of naïve and healthy individuals and in vaccinated COVID-19 patients.

2) There is no sufficient information for human subjects. It is essential to have the information of human subjects as a part of the methods. What were the underlying conditions for each patient? Were the doses of dexamethasone therapy in all patients

the same or different? Besides dexamethasone, were there other therapies that potentially affect immune response? It seems that non-COVID controls were from a different study. The nature of these controls should also be included.

Response

We updated the supplementary data (Supplementary Table 1) with additional information detailing underlying health conditions. We also added the dexamethasone doses used.

We also included an additional experimental group of naïve healthy individuals that received the first dose of BNT162b.

In the original analysis we had used healthy individuals from our previous study (PMID: 33608566) as the controls. In the revision, we compared the group of five unvaccinated and four vaccinated patients, all of which had been infected with the Beta variant. We also compared four vaccinated Beta-infected patients with eight vaccinated healthy individuals.

3) The GSEA was a good starting point but further analyses are required to understand how individual pathways were enhanced or suppressed. It was unclear how virus infection related genes (Fig. 1c) or vaccine related immune genes (Fig. 2c) were defined?

Response

We have now added additional, more detailed analyses. Specifically, we have focused on the interferon-JAK-STAT pathway and identified key genes, such as OAS members, *DHX58* and the *IFIT* family, involved in the innate immune system and antiviral response. We have analyzed their regulation by interferons using primary cells through RNA-seq and CHIP-seq analyses (Figures 2 and 3). We have also investigated the *OAS1* haplotype status (Supplementary Table 6) of the 'Neanderthal' mutation that had been linked to milder COVID-19. We identified three patients homozygous for this mutation but there was no apparent impact in the disease severity.

We have also investigated whether the genes hyperactivated in vaccinated patients are reminiscent to a vaccine response in naïve healthy individuals. For this purpose, we conducted additional RNA-seq on a group of eight naïve individuals at the time of vaccination and 7-10 days later (Figure 5).

4) In Fig 4, there was a separation between the elderly and younger samples. I don't know why such analysis supports the conclusion in the abstract "the transcriptomes of the older vaccinated group ... more similar to the immune response of younger unvaccinated individuals".

Response

In this revision we combined all unvaccinated hospitalized patients ($n=5$) which gave us sufficient power to conduct this study and draw sound conclusions. We do not distinguish between young and old.

5) There are much inaccurate and missing information throughout the manuscript. For example.

Line 74 “Four patients had received the first dose of the BNT162b (Pfizer-BioNTech) vaccine 11 days before the onset of COVID-19 symptoms and one patient eight days prior (Fig. 1a).” It should be “Three... and one”

Fig. 1b, 2a, 4a should have labelled subject ID

Fig 2e, X-axis is not labelled.

Fig 4a, Better labelled as old vs middle-aged, 48-62 yr old are not “young”

Response

We carefully edited the manuscript.

Reviewers' comments:

Reviewer #1 (Remarks to the Author):

In this revised manuscript by Lenninghausen and colleagues, the authors substantially modify their analyses and conclusions to focus on the interferon-JAK-STAT regulated pathways that are induced by vaccination and/or SARS-CoV-2 infection. They combined data from 4 vaccinated COVID-19 patients and 5 unvaccinated patients and looked at the immune transcriptomes with a focus on JAK-STAT-mediated signaling. They identified increased expression of particular genes, such as oligoadenylate synthetase (OAS1), and showed that this expression was also elevated in naïve individuals after receiving their first dose of the BNT162b vaccine.

The strengths of the study remain the same, that added knowledge on immune transcriptome pathways in COVID-19 infection and vaccination is lacking and is very important. I also appreciate the new data on naïve vaccinated individuals that were available for comparison. The key weaknesses of very few patients still holds. More worrisome to me, however, is that the authors completely overhauled their analyses such that this review is akin to reviewing a completely new study. Although some of my comments were handled, the new analyses raise a few additional questions that in my opinion should be addressed, as follows:

(1) It is still unclear to me why the PCA plots in the original submission were so skewed. I don't think that the authors should simply remove those plots because the PCA plots are a powerful tool to determine whether there is sampling bias or batch effect. Any batch effect would be even more pronounced given the low numbers. The authors should provide the PCA plots as supplementary data along with an interpretation, or generate other equivalent plots to assess for batch effect, such as multidimensional data sets (MDS) analysis.

(2) In the ranking of pathways, I noticed that p53 cell cycle arrest was also significantly differentially expressed. Did the authors look at p53-regulated pathways? These pathways are also highly thought to be significant in coronavirus infection and also related to STAT activation (see Cardozo and Hainaut, <https://www.ncbi.nlm.nih.gov/pmc/articles/PMC7924916/>; Ramaiah, <https://www.ncbi.nlm.nih.gov/pmc/articles/PMC7324924/>; Matsuyama, et al.. Given that OAS-related pathways were detected by the authors in the naïve vaccinated individuals, it would be interesting to see whether p53-regulated pathways are also detected, as I would assume that they would be more specific for acute infection/inflammation as seen in infected COVID-19 patients.

(3) Was elevated ACE2 expression seen in the naïve vaccinated individuals?

(4) The authors should provide a figure or data showing exactly how much overlap there was between the gene expression of naïve vaccinated individuals and vaccinated/unvaccinated COVID-19 patients. Can they provide a Venn diagram?

(5) I am still unclear on what controls were used for the study to evaluate differential gene expression. Were the controls all the same individual at time $t=0$ prior to vaccination? What were the controls used for the hospitalized patients and how many controls were used? The authors state that "In the original analysis we had used healthy individuals from our previous study (PMID: 33608566) as the controls. In the revision, we compared the group of five unvaccinated and four vaccinated patients, all of which had been infected with the Beta variant. We also compared four

vaccinated Beta-infected patients with eight vaccinated healthy individuals that had enrolled in our vaccination study." They don't mention what controls were used when "comparing the group of 5 unvaccinated and 4 vaccinated patients" or when "comparing 4 vaccinated Beta-infected patients with eight vaccinated healthy individuals that had enrolled in our vaccination study"?

Reviewer #2 (Remarks to the Author):

The revised version is much improved, not only fully addressing my concerns but also provide new analysis. The focus on antiviral mechanism enhanced by vaccine is intriguing.

I only have a minor suggestion for the title. "... enhances interferon-JAK-STAT regulated genetic programs ..." is not entirely clear for me. Would it be more straightforward as "... enhances interferon-JAK-STAT regulated antiviral programs ..."?

Response to the reviewers

Reviewer #1

(1) It is still unclear to me why the PCA plots in the original submission were so skewed. I don't think that the authors should simply remove those plots because the PCA plots are a powerful tool to determine whether there is sampling bias or batch effect. Any batch effect would be even more pronounced given the low numbers. The authors should provide the PCA plots as supplementary data along with an interpretation or generate other equivalent plots to assess for batch effect, such as multidimensional data sets (MDS) analysis.

Response

We used the *RUV* method in the *RUVSeq* package to detect the hidden batch effects and corrected unwanted variation present in the data. Residuals were normalized using *RUV* and PCA plots were generated. As requested by the reviewer, we now present PCA and RLE analyses in Supplementary Figure 4.

(2) In the ranking of pathways, I noticed that p53 cell cycle arrest was also significantly differentially expressed. Did the authors look at p53-regulated pathways? These pathways are also highly thought to be significant in coronavirus infection and also related to STAT activation (see Cardozo and Hainaut, <https://www.ncbi.nlm.nih.gov/pmc/articles/PMC7924916/>; Ramaiah, <https://www.ncbi.nlm.nih.gov/pmc/articles/PMC7324924/>; Matsuyama, et al.. Given that OAS-related pathways were detected by the authors in the naïve vaccinated individuals, it would be interesting to see whether p53-regulated pathways are also detected, as I would assume that they would be more specific for acute infection/inflammation as seen in infected COVID-19 patients.

Response

As the reviewer noted we identified enrichment of p53 cell cycle arrest (Fig. 2a) and have now added presentation of significantly regulated p53 pathway genes (Fig. 2i) with relevant discussion and additional references (page 6).

“In coronavirus infection, similar to other viruses, p53 has been shown to play a role in limiting viral replication.^{24,25} p53 degradation triggered by coronavirus nsp3 protein is suggested to contribute to virus survival. We found that vaccinated patients demonstrated enrichment in p53 regulated cell cycle arrest genes, specifically *CCNA1*, *E2F8* and *E2F1* (Fig. 2a and i), consistent with a possibility of higher p53 activity levels in the vaccinated patients. Interestingly *E2F8*, but not *CCNA1* or *E2F1*, was modestly up-regulated seven days post-vaccination in the naïve vaccinated individuals (1.5-fold, *padj*=0.04).”

(3) Was elevated ACE2 expression seen in the naïve vaccinated individuals?

Response

We analyzed ACE2 expression in our cohorts and observed increased expression in the hospitalized patients regardless of the vaccination status, as compared to the healthy naïve population. This data has been added (Supplementary Fig. 2) with relevant discussion and additional references (pages 7-8).

“The *ACE2* gene, encoding the receptor of SARS-CoV-2, is regulated by interferons through the JAK-STAT pathway^{12,31} opening the possibility of increased expression in hospitalized COVID-19 patients. Notably, *ACE2* mRNA levels significantly elevated in hospitalized patients regardless of their vaccination status as compared to healthy volunteers receiving the first dose of BNT162b (Supplementary Fig. 2). The *ACE2* gene has two distinct promoters, one controlling the full-length mRNA and an intronic one responsible for the short form of *ACE2*^{12,32-34}. SARS-CoV-2 can infect immune cells³⁵ and an increase of the native *ACE2* would elevate the propensity of infection.”

(4) The authors should provide a figure or data showing exactly how much overlap there was between the gene expression of naïve vaccinated individuals and vaccinated/unvaccinated COVID-19 patients. Can they provide a Venn diagram?

Response

A Venn diagram with genes significantly regulated between Day0 and Day7 in naïve vaccinated individuals and Day0 vs 1st time point of vaccinated/unvaccinated COVID-19 patients has been added (Supplementary Figure 3) with description in the text (page 8).

“The number of overlapping genes significantly regulated between the gene expression of naïve vaccinated individuals and vaccinated/unvaccinated COVID-19 patients are summarized in Supplementary Fig. 3.”

(5) I am still unclear on what controls were used for the study to evaluate differential gene expression. Were the controls all the same individual at time t=0 prior to vaccination? What were the controls used for the hospitalized patients and how many controls were used? The authors state that “In the original analysis we had used healthy individuals from our previous study (PMID: 33608566) as the controls. In the revision, we compared the group of five unvaccinated and four vaccinated patients, all of which had been infected with the Beta variant. We also compared four vaccinated Beta-infected patients with eight vaccinated healthy individuals that had enrolled in our vaccination study.” They don’t mention what controls were used when “comparing the group of 5 unvaccinated and 4 vaccinated patients” or when “comparing 4 vaccinated Beta-infected patients with eight vaccinated healthy individuals that had enrolled in our vaccination study”?

Response

In response to reviewer comments of the initially submitted version (May 9, 2021), aspects of the analysis were revised and no data from the healthy individuals from our

previous study (PMID: 33608566) were included as controls in the revision submitted in August. Instead, the controls used for this revision were the eight healthy naïve volunteers (age 27-51 yrs.) participating in our vaccine study. Samples before and after vaccination were derived from the same individuals. As a control to the hospitalized patients, we used Day0 data of the healthy naïve volunteers receiving a single vaccination and compared them to the samples from the first time point of unvaccinated/vaccinated COVID-19 patients. Detailed subject and sample information is included in Supplementary Data1.

Reviewer #2:

The revised version is much improved, not only fully addressing my concerns but also provide new analysis. The focus on antiviral mechanism enhanced by vaccine is intriguing.

I only have a minor suggestion for the title. "... enhances interferon-JAK-STAT regulated genetic programs ..." is not entirely clear for me. Would it be more straightforward as "... enhances interferon-JAK-STAT regulated antiviral programs ..."?

Response

We thank the reviewer for the suggestion to further clarify the title, which we happily accept.

REVIEWERS' COMMENTS:

Reviewer #1 (Remarks to the Author):

In this improved revised manuscript, the authors have addressed the remaining concerns that I raised in their rebuttal. I am in favor of accepting the manuscript for publication. Great job!